# Potential of Selected African Medicinal Plants as Alternative Therapeutics against Multi-Drug-Resistant Bacteria

**DOI:** 10.3390/biomedicines11102605

**Published:** 2023-09-22

**Authors:** Bertha N. Moiketsi, Katlego P. P. Makale, Gaolathe Rantong, Teddie O. Rahube, Abdullah Makhzoum

**Affiliations:** Department of Biological Sciences and Biotechnology, Faculty of Science, Botswana International University of Science and Technology (BIUST), Private Bag 16, Palapye, Botswana; mb21100082@studentmail.biust.ac.bw (B.N.M.); mk21100078@studentmail.biust.ac.bw (K.P.P.M.); rantongg@biust.ac.bw (G.R.)

**Keywords:** antimicrobial resistance, antibiotics, bioactive compounds, secondary metabolites, indigenous plants, Africa

## Abstract

Antimicrobial resistance is considered a “One-Health” problem, impacting humans, animals, and the environment. The problem of the rapid development and spread of bacteria resistant to multiple antibiotics is a rising global health threat affecting both rich and poor nations. Low- and middle-income countries are at highest risk, in part due to the lack of innovative research on the surveillance and discovery of novel therapeutic options. Fast and effective drug discovery is crucial towards combatting antimicrobial resistance and reducing the burden of infectious diseases. African medicinal plants have been used for millennia in folk medicine to cure many diseases and ailments. Over 10% of the Southern African vegetation is applied in traditional medicine, with over 15 species being partially or fully commercialized. These include the genera *Euclea*, *Ficus*, *Aloe*, *Lippia.* And *Artemisia*, amongst many others. Bioactive compounds from indigenous medicinal plants, alone or in combination with existing antimicrobials, offer promising solutions towards overcoming multi-drug resistance. Secondary metabolites have different mechanisms and modes of action against bacteria, such as the inhibition and disruption of cell wall synthesis; inhibition of DNA replication and ATP synthesis; inhibition of quorum sensing; inhibition of AHL or oligopeptide signal generation, broadcasting, and reception; inhibition of the formation of biofilm; disruption of pathogenicity activities; and generation of reactive oxygen species. The aim of this review is to highlight some promising traditional medicinal plants found in Africa and provide insights into their secondary metabolites as alternative options in antibiotic therapy against multi-drug-resistant bacteria. Additionally, synergism between plant secondary metabolites and antibiotics has been discussed.

## 1. Introduction

Antimicrobial resistance (AMR) is a well-recognized global health problem, affecting both rich and poor nations [1]. AMR is the ability of microorganisms to survive the effects of antimicrobials such as antifungals and antibiotics. Many pathogenic bacteria have developed resistance to almost all available antibiotics, and even newly developed antibiotics will ultimately become ineffective against the continuously evolving multi-drug-resistant (MDR) bacteria [1,2,3]. Some examples of clinically important MDR bacteria are Vancomycin-resistant *Enterococci* (VRE), Carbapenem-resistant *Acinetobacter baumannii* (CRAB), Carbapenem-resistant *Enterobacteriales* (CRE), XDR (extensively drug-resistant) *Pseudomonas aeruginosa*, extended-spectrum β-lactamase- (ESBL)-producing *Enterobacteriales*, and methicillin-resistant *Staphylococcus aureus* (MRSA) [4,5,6]. The *E. faecium*, *S. aureus*, *K. pneumoniae*, *A. baumannii*, *P. aeruginosa*, *Enterobacter* species, and *E. coli* (ESKAPEE) acronym is commonly used to refer to highly virulent bacterial pathogens known to “escape” treatment from multiple antibiotics and other traditional treatments [7,8]. The rapid development and spread of MDR bacteria is a rising One-Health challenge, impacting humans, animals, and the environment, which means that MDR bacteria and the associated AMR genes are capable of circulating among different habitats, making them difficult to control [4,9]. It has been estimated that 10 million deaths will be a result of AMR by the year 2050, and the World Health Organization (WHO) predicts over 24 million people globally will be condemned to poverty as result of the AMR burden [2]. The European Centre for Disease Prevention and Control (ECDC) estimates that today AMR is already responsible for ca. 25,000 deaths and €1.5 billion in health expenditures per year in Europe alone, and given the limitations of the availability of data, the death toll is estimated to be significantly higher in Africa, where the expenditure is much higher than in the first-world countries [1]. In fact, the Africa Centres for Disease Prevention and Control (CDC) has urged the public, academic institutions, farmers, veterinarians, and medical and professional organizations to become ‘antibiotic guardians’ by cutting the unnecessary use of antibiotics in order to slow resistance [10,11]. Some bacteria are naturally resistant to antibiotics via mutation and horizontal gene transfer; however, the frequent use of antibiotics to treat bacterial infections in healthcare, coupled with anthropogenic activities such as wastewater reuse, among other agricultural practices, contributes to the rapid spread of antibiotic resistance [2,7].

In low- and middle-income countries (LMICs), there is lack of innovative research on antibiotic usage, surveillance, and the discovery of novel therapeutic options; therefore, fast and effective drug discovery is crucial to help reduce the rise of deadly infections, which are now contributing to more deaths in these countries [12]. Thus, novel efficient antibacterial agents and alternative strategies are urgently required to fill the void of antibiotic discovery and development [13]. Medicinal plants have long been used as medicines to treat microbial infections, among other human diseases; hence, their antibiotic development should now be focused on, as they have proved to be the best alternatives for novel antibacterial targets and can be effective against MDR bacteria [3]. Compared to synthetic chemotherapeutic drugs, natural antibacterial agents and their analogues still dominate the multiple classes of antibiotics, such as β-lactams, tetracyclines, aminoglycosides, and polypeptides, which are routinely used in healthcare [13]. These natural antibacterial agents possess advantages in structural and chemical diversity, accessibility, robust activity, and peculiar modes of action, and present lower health risks associated with human toxicity or side effects [10,14,15]. Additionally, plants have a superior ability to assimilate genetic information and produce complex molecules that can be used to make more effective therapeutics. Moreover, there are significantly lower facility and production costs associated with plant-made drugs, as it costs significantly less to grow plants and mass-produce pharmaceutical compounds, which can allow more capital to be invested into the research and development of new therapeutics. Research focusing on plant secondary metabolites and their possible effectiveness against antibiotic-resistant bacteria could lead to much anticipated discoveries in terms of drug development.

In this paper, we highlight some promising traditional medicinal plants found in the African continent, review the current literature and studies on the therapeutic potential of traditional medicinal plants, and provide insights into their secondary metabolites and modes of action as alternative options in antibiotic therapy against MDR bacteria.

## 2. Materials and Methods

A comprehensive literature search was conducted on MDR bacteria of clinical relevance, along with their associated resistance mechanisms. A total of 36 selected medicinal plants indigenous to the continent of Africa were selected (refer to Figure 1). They were also reviewed together with their associated secondary metabolites known to have potential antibacterial properties and their modes of action. The reviewed literature included conference papers, books, theses, and papers published in peer-reviewed international journals, as well as reports from international, regional, and national organizations. The PubMed and Scopus databases, as well as the Google Scholar search engine, were used in mining the literature using the following keywords alone or in combination: muti-drug-resistant bacteria; plant bioactive compounds; plant secondary metabolites; indigenous plants; Africa. There was no restriction on the year of publication of the selected literature; the information in this article was collected in publications from both original and review articles spanning from 1994 to 2023.

## 3. Results and Discussion

### 3.1. Clinically Important Multi-Drug-Resistant Bacteria and Modes of Antibiotic Resistance

Antibiotic resistance is found in both Gram-negative and Gram-positive strains of bacteria, which are leading causes of hospital- and community-acquired infections, ranging from common infections such as skin and soft tissue infections to life-threatening infections [4]. MDR bacterial infections account for millions of global deaths annually, with over 40% being neonatal deaths and with that percentage expected to increase in the absence of effective therapeutic drugs [4]. Gram-negative bacteria belonging to the order *Enterobacteriales* have since developed resistance mechanisms posing a serious threat to human health, especially in hospitals and nursing homes [4,7,16]. *Enterobacter* species are also Gram-negative and are characterized as facultatively anaerobic, rod-shaped bacteria of the *Enterobacteriales* family, which includes *E. coli* and *K. pneumoniae* [17]. This group of pathogens are a major cause of urinary and respiratory tract infections, causing bacteremia and pneumonia in the immunocompromised [17,18]. *Acinetobacter* species (e.g., *A. baumannii*) are Gram-negative, aerobic, non-fermenting, non-fastidious, ubiquitous coccobacillus, or pleomorphic bacteria, and are responsible for bloodstream infections and ventilator-associated pneumonia [17,18]. *Acinetobacter* species have the ability to resist desiccation and form biofilms, and the presence of fundamental virulence factors, such as secretion systems, surface adhesins, and glycoconjugates, aggravate their pathogenicity [19,20]. *Pseudomonas aureginosa*, is an example of another Gram-negative bacterium that causes urinary tract infections, surgical site infections, pneumonia, septicemia, and bacteremia, especially in immunocompromised individuals [21]. *P. aeruginosa* displays innate resistance to a wide array of antibiotics. It is resistant to a wide spectrum of antibiotic classes, including penems and β-lactam, and its resistance to fluroquinolones is due to its ability to mutate on DNA gyrase or topoisomerase [21,22]. *P. aeruginosa* employs a variety of mechanisms such as alterations in porin channels, efflux pumps, targets modifications, and β-lactamases to exert resistance to antimicrobial agents [21,22,23]. *A. baumannii*, *P. aeruginosa*, and several members of the *Enterobacteriales* family exhibit broad resistance to carbapenems antibiotics; hence, at the top of the WHO priority list is research and development for new antibiotics, which are urgently needed [17,24].

The Gram-positive bacteria of clinical concern include the genera Bacillus (e.g., Enterococcus species, Staphylococcus aureus), Clostridium (*C. botulinum*, *C. perfringens*), Listeria (*L. monocytogenes*), Gardenella (*G. vaginalis*), and Corynebacterium (*C. diphtheriae*) [24]. Enterococcus faecalis is an enterococcal bacterium that is responsible for infections in the gut of humans, and is known to cause severe infections in immunocompromised individuals [13,16]. VRE readily accumulate mutations and exogenous genes (VanA, VanB, VanD, VanE, VanG, VanL) that confer resistance to vancomycin, including other antibiotics classes such as β-lactam [25,26]. *S. aureus* is a Gram-positive spherical bacterium that is normal microflora of the skin and the nasal mucosa, which can also be pathogenic [27,28]. Pathogenic strains of *S. aureus* normally cause life-threatening soft tissue abscesses, pneumonia, septicemia, and bacteremia, and can cause infections from contaminated medical implants. Its ability to form biofilms also poses a challenge in antibiotics-mediated treatments [29]. Additionally, due to the secretion of the TSST-1 exotoxin in some strains, *S. aureus* can also cause toxic shock syndrome [30,31]. *S. aureus* has evolved to develop resistance to vancomycin, methicillin, and many β-lactam classes of antibiotics [31]. MRSA harbors a mecA gene on the staphylococcal cassette chromosome mec (SCCmec) and codes for PBP2a [32]. A protective protein bound to the ribosomes of the bacterial cell inactivates the antibiotics via ‘target alteration’ by altering their structural confirmation [33].

The general mechanisms of antibacterial resistance range from alterations of binding sites, alterations of the bacterial porins’ structure, antibiotics efflux through the bacterial efflux pump structure, and destruction of antibacterial agents by hydrolytic enzymes [5,16]. Additionally, MDR bacteria can resist antibiotics via one mechanism or by combining more than one to produce their multiple resistance to antibiotics and other antimicrobials, including disinfectants and heavy metals in personal care products [7]. The examples of clinically relevant bacteria and their modes of action are summarized in Table 1.

### 3.2. Diversity and Distribution of African Medicinal Plants with Potential Antimicrobial Properties

Thirty-six promising medicinal plant species are highlighted in this review. They are widely distributed across the African region. Southern Africa has the majority, followed by East Africa, then Central Africa, while West Africa and North Africa have the least (Figure 1). *Hibiscus calyphyllus* is common to all the regions; *Cassia abbreviata*, *Dicoma anomala Sond*, and *Securidosa longipendunculata* are also found in all the regions except North Africa. *Dichrostachys cinerea* is also universal to all regions except West Africa. Fifteen species (*Adansonia digitata*, *Aloe zebrina Baker*, *Aloe ferox*, *Artemisia afra*, *Boscia albitrunca*, *Colophospermum mopane*, *Combretum hereroense*, *Commiphora glandulosa*, *Cynodon transvaalensis*, *Euclea undulata*, *Harpagophytum procumbens*, *Hirpicium bechuanense*, *Lippia javanica*, *Ozoroa paniculosa*, *Sanseviera scabrifolia*, *Sclerocarya birrea*) were found to be unique to the Southern African region, nine (*Elephantorhiza goetzei*, *Grewia bicolor*, *Grewia flava*, *Harpagophytum procumbens*, *Lippia scaberrima*, *Mimusopus zeyheri*, *Myrothamnus flabellifolius*, *Scadoxus puniceus*, *Terminalia sericea*) are common to Southern and East Africa, and two (*Capparis tomentosa*, *Laphangium luteoalbum*) are found only in Southern and North Africa. *Asparagus africanus*, *Vanguera infausta*, *Ximenia americana*, and *Ximenia caffra* were found to be distributed in Central, East, and Southern Africa.

Over 10% of the Southern African vegetation is applied in traditional medicine, with over 15 species being partially or fully commercialized, which can be found in local pharmacies. These include *Hibiscus calyphyllus*, *Harpagophytum procumbens*, *Cassia abbreviata*, *Aloe ferox*, *Lippia javanica*, and *Artemisia afra*, amongst many others (Figure 1) [35,36,37,38,39,40]. *C. abbreviata*, which is also known as long-tail *Cassia* (or Monepenepe in Setswana, a native language of Botswana), belongs to the *Caesalpiniaceae* family and is characterized by thick bushes, brown bark, a rounded crown, yellowish leaves, and sweet-scented flowers, as well as long cylindrical dark brown fruits hanging in pods (Figure 1) [41]. The sun-dried bark is boiled in water and served as a hot tea to individuals with miscellaneous stomach ailments, skin problems, and STIs [38] (Table 2). *H. calyphyllus*, described as a large yellow hibiscus (or Motsididi in Setswana), is a leafy shrub with wide and simple serrate leaves and yellow flowers with a dark red center (Figure 1), belonging to the *Malvaceae* family [40]. The flowers, which have been reported to be rich in flavonoids and phenolic acids, are traditionally sun-dried, boiled, and served as a hot beverage to treat intestinal ailments in many sub-tropical parts of Africa [40,42] (Table 2). *A. afra*, also known an African wormwood, belongs to the *Asteraceae* family [43]. It is an erect, perennial woody shrub with oval-shaped, greyish-looking leaves (Figure 1). The leaves, stems, and roots are rich in terpenoids, tannins, saponins, and glycosides, which are active against colds, coughs, influenza, sore throat, malaria, asthma, pneumonia, and diabetes [44]. These parts of the plants are served pulverized as a hot beverage [45] (Table 1).

*L. javanica* (lemon bush), of the family *Verbenaceae*, is a woody shrub with aromatic leaves that gives a lemon-like smell, which is used as a culinary spice, as well as to treat coughs, colds, fever, chest ailments, kidney stones, measles, rashes, and stomach problems [37,46]. Small, dense spikes of white flowers are borne in the axils of leaves (Figure 1). Dried lemon bush leaves are boiled and consumed as is or applied on affected areas [47].

*H. procumbens* of the sesame seed or *Pedaliaceae* family, popularly known as devil’s claw, is rich in terpenoids, iridoid glycosides, glycosides, and acetylated phenolic compounds [47]. It is a tuberous perennial plant with creeping stems and dark pink flowers (Figure 1). Devil’s claw is used for a wide variety of health conditions in the form of hot or cooled infusions, decoctions, tinctures, powders, and extracts to treat blood diseases, urinary tract infections, postpartum pains, sprains, sores, sexually transmitted diseases, ulcers, and boils [48]. Commercially, the secondary tubers or roots are pulverized into capsules [47] (Table 2).

*Aloe ferox* (*Xanthorrheaceae*, previously *Asphodelaceae*, *Aloaceae*, or *Liliaceae*; commonly known as the bitter aloe in English and kgwaphane or mokhwapha in Setswana) is a cherished, popular, ornamental single-stemmed plant with erect racemes of red, orange, yellow, or rarely white flowers, with spreading or gracefully curved thorny leaves (Figure 1) [49,50]). Traditionally, the fresh leaf is cut up, the flesh is extracted and directly applied on the affected areas, and it is consumed as is or diluted in cold water [39]. Commercially, it is incorporated into different cosmetic products, health drinks, foods, and beverages to deal with various ailments [50] (Table 2). *B. albitrunca* is a medium-sized evergreen tree belonging to the *Capparaceae* or *Caper* family and is served as hot coffee or tea [51]. The bark, leaves, and roots are mainly used as herbal medicines for STIs and skin and stomach infections [51]. In fact, in a study by Pendota et al. in 2015, crude, dichloromethane, ethyl acetate, and butanol leaf extracts were evaluated and confirmed for antibacterial activities against *B. subtilis*, *S. aureus*, *E. coli*, and *K. pneumoniae*. Motlhanka et al. conducted a study to assess the antibacterial properties from the resin of *C. glandulosa*. This is a single-stemmed tree with greyish-green to yellowish-green flaking bark that belongs to the family *Burseraceae* [52]. Crude aqueous and chloroform extracts of the stem resin, as well as the isolated compound, exhibited good in vitro antibacterial activity against Gram-positive bacteria, *B. subtilis*, *C. perfringens*, and *S. aureus*, as well as multi-drug-resistant *S. aureus*, XU212-tetracycline-resistant, and SA1199B-norfloxacin-resistant strains [47].

*X. caffra* (*Ximeniaceae*), commonly known as “sour plum”, is traditionally used both topically and orally to treat a wide range of bacterial infections such as wounds, STIs, respiratory ailments, digestive tract ailments, colds, and coughs [53,54]. Phytochemical investigations of the bark, fruits, leaves, roots, and seeds of the sour plum revealed various compounds, including flavonoids, phenols, phytosterols, and tannins as active compounds against bacterial pathogens [55]. The methanol extracts of *X. caffra* roots exhibited antibacterial activities against *S. aureus* and *S. epidermidis* [56]. *D. cinerea* is a thornbush belonging to the Leguminosae subfamily Mimosoideae. It is a medicinal plant that is native to Africa and rich in tannins in its leaves, bark, and roots [57,58]. Tannins were isolated from *D. cinerea* and assayed against *S. aureus*, *S. boydii*, *S. flexneri*, *E. coli*, and *P. aeruginosa* using the agar diffusion method [57]. The associated tannins exhibited antibacterial activities against all test microorganisms. This explains why the dried bark, roots, and leaves are served as a hot tea to traditionally treat sexually transmitted, respiratory, dental, skin, and intestinal infections [58].

In a study aimed at investigating the in vitro antimicrobial activity of ethanolic extracts of seventeen species of *Sansevieria*, including *S. scabrifolia*, against *E. coli* using the agar disk diffusion method, a degree of inhibition was found [35]. The leaves of this species are used to treat ear infections, toothache, and diarrhea [54].

*C. tomentosa* (belonging to the family *Capparaceae*) is a scrambling shrub that grows as high as 10 m tall and is found across North Africa and Southern Africa (Table 2). It is used to treat pneumonia, coughs, headaches, tuberculosis, and gonorrhea [59]. The associated phytochemicals that are extracted from the hairy yellow-green twigs and leaves are linked to its unique biological, bactericidal, and bacteriostatic activities, which include alkaloids, L’stachydrine, saponin glycosides, phytosterols, terpenoids, tannins, and anthranoids [59]. Studies have shown that this species has good antimicrobial activity against antibiotic-resistant *S. aureus*, *S. pyogenes*, *E. coli*, and *P. aeruginosa* [60]. *O. paniculosa* (*Anacardiaceae*) is an evergreen, semideciduous, small- to medium-sized single-stemmed tree that is rich with phenols [61]. Phenolic-enriched leaf extracts of *O. paniculosa* were prepared using a mixture of 1% HCl-acidified 70% acetone and n-hexane, and then tested against *S. aureus*, *P. aeruginosa*, *E. coli*, and *E. faecalis* [61]. These extracts had good activities relating to diarrhea mechanisms or pharmacological relevance. *E. undulata* (belonging to the *Ebenaceae* or *Ebony* family) has egg-shaped to wide, bluntly pointed waxy leaves, yellowish fragrant flowers, and globose fleshy fruits that are all traditionally used for the treatment of body pains, chest complaints, cough, diarrhea, headaches, heart disease, and tooth aches because of their wealth of diterpenes, flavonoids, naphthoquinones, phytosterols, saponins, and tannins [62]. In a previous study, the antimicrobial activity of *E. undulata* chewing sticks against multi-drug-resistant S. mutans was determined [63]. The minimum inhibitory concentrations ranged from 0.385 to 11.22 mg/mL and the minimum bactericidal concentrations from 0.485 to 20.20 mg/mL. *T. sericea* (of the family *Combretaceae*) is a small to medium deciduous rounded flowering shrub whose roots are traditionally used to treat diarrhea, skin rashes, tuberculosis, and opportunistic infections associated with HIV/AIDS in Botswana [64]. It has been reported that dichloromethane/methanol (1:1) extracts of the stems, bark, leaves, and roots have antibacterial activity against *B. subtilis*, *B. cereus*, *S. aureus*, *E. coli*, *K. pneumoniae*, *P. aeruginosa*, *S. sonnei*, *S. typhimurium*, and *S. epidermidis* [65]. The compounds isolated from this species so far include a triterpene sericoside, resveratrol-3-*O*-β-D-rutinoside, and hydroxystilbene glycoside [66].

*S. scabrifolia* (also known as Mosokelatsebeng in Setswana) is a stemless evergreen perennial succulent that grows from a thick rhizome. Its fleshy leaves are warmed in a fire and the juice is squeezed into the ear or tooth to treat ear infections and cavities in Botswana [54]. In Namibia, the leaf sap is applied to wounds to prevent infection and accelerate healing [67]. This bactericidal capacity was also confirmed by Tkachenko et al., showing that the crude extracts had antibacterial activity against pathogenic *E. coli* [67]. The antibacterial activity in the *Sansevieria* genus may be due to the presence of alkaloids, saponins, terpenoids, steroids, glycosides, and tannins [65].

**Table 2 biomedicines-11-02605-t002:** Summary of African medicinal plants, their uses in folk medicine, and their geographical locations.

Scientific Name	Setswana Name (Common Name)	Manner of Administration	Ailments Treated	Distribution in Africa	References
*Adansonia digitata*	Mowana (African baobab)	Fruit is added to milk and consumed as is. Seeds are crushed and the oil is extracted. Bark, flowers, leaves, and roots are pulverized and consumed as a hot beverage	Intestinal infections, respiratory infections, skin infections	Mozambique	[68]
				Namibia	
				South Africa	
*Aloe zebrina Baker/Aloe ferox*	Kgophane/Mokhgwapha (variegated aloe)	Leaf flesh is cut up and juice extracted to be applied as is or diluted in water	Skin infections, STIs	Angola	[39]
				Namibia	
				Mozambique	
				Zambia	
				Zimbabwe	
*Asparagus africanus*	Mhalatsamaru/wild Asparagus	Dried tubers or roots are boiled and served as a beverage	Genital sores and wounds	Eswatini	[47]
				Lesotho	
				Mozambique	
				Namibia	
				Zimbabwe	
*Artemisia afra*	Lengana (African woodworm)	Leaves, stems, and roots are boiled in water and served hot	Respiratory infections and related symptoms such as fevers	Cameroon	[45]
				Chad	
				Ethiopia	
				Kenya	
				Namibia	
				Tanzania	
				Uganda	
				South Africa	
				Zimbabwe	
*Boscia albitrunca*	Motopi (Shepherd’s tree)	Pulverized leaves are served in warm milk	Respiratory infections	Eswatini	[69]
				Mozambique	
				South Africa	
				Zambia	
				Zimbabwe	
*Capparis tomentosa*	Motawana (African caper)	Dried bark, stems, leaves, or roots are boiled and consumed as tea	Respiratory system infections, STIs	Tropical Africa	[38]
				Namibia	
				Mozambique	
				South Africa	
*Cassia abbreviata*	Monepenepe (long pod cassia)	Sun dried bark or stem is boiled in water and served hot	Diarrhea, skin diseases, STIs, stomach infections	DRC	[38]
				Eswatini	
				Kenya	
				Mozambique	
				Namibia	
				Somalia	
				South Africa	
				Tanzania	
				Zambia	
				Zimbabwe	
*Colophospermum mopane*	Mophane (mopane tree)	Pulverized seeds are boiled and orally administered Gum from the stem is applied on affected areas. Roots are boiled to make a hot beverage	Stomach infections, STIs, sore eyes	Angola	[52]
				Malawi	
				Mozambique	
				Namibia	
				Zambia	
				Zimbabwe	
*Combretum hereroense*	Mokabi (russet bushwillow)	Fruits are eaten as is. Dried leaves are boiled to make tea	Stomach infections	Eswatini	[47]
				Mozambique	
				Namibia	
				South Africa	
				Zimbabwe	
*Commiphora glandulosa*	Moroka (tall common corkwood)	Several incisions are made to the plant during winter, and the resinous exudate is harvested and applied to affected areas. The bark is boiled and consumed as tea	Skin and soft tissue infections	Angola	[70]
				Mozambique	
				Namibia	
				South Africa	
				Zambia	
				Zimbabwe	
*Cynodon transvaalensis*	Monamane (Burtt-Davy)	Ground roots and root bark are sun-dried and pulverized for beverage preparation	Stomach infections, fever	Angola	[47]
				Eswatini	
				Lesotho	
				Malawi	
				Mozambique	
				Namibia	
				South Africa	
				Zambia	
				Zimbabwe	
*Dichrostachys cinerea*	Moselesele (sickle bush)	Dried bark, roots, and leaves are boiled to make the beverage. Seeds are crushed to extract the oil. Incisions on the stem are made to collect gum	STIs, intestinal infections, dental infections, skin infections	From Eastern Cape RSA to Tropical East Africa	[57]
*Dicoma anomala Sond.*	Tlhonya (fever/stomach bush	Dried roots are boiled in water and served as tea	Stomach infections and diarrhea	Angola	[71]
				Burundi	
				DRC	
				Rwanda	
				South Africa	
				Tanzania	
				Zambia	
				Zimbabwe	
*Elephantorhiza goetzei*	Mositsane/Mosidi (large been elephant root)	Bark and root decoctions are taken orally	Intestinal infections, urinary tract infections, STIs	Malawi	[72]
				Mozambique	
				Namibia	
				South Africa	
				Tanzania	
				Zambia	
				Zimbabwe	
*Euclea undulata*	Motlhakola (common guarri)	The roots directly brush the teeth and mouth. Pulverized leaves make a hot beverage for oral ingestion. Bark is boiled and served as hot tea	Mouth sores and wounds, tooth ache and sore throat, intestinal infections	Namibia	[62]
				Swaziland	
				Zimbabwe	
*Grewia bicolor; Grewia flava; Grewia flavescens*	Moretlwa/Mogwana (wild currant/velvet raisins/brandy bush)	Fruit is eaten or prepared into a dry pulp. Fresh or dry roots are boiled to prepare a beverage	Cavities Urinary tract infections, skin infections	Angola	[73]
				Eswatini	
				Ethiopia	
				Namibia	
				Zimbabwe	
*Harpagophytum procumbens*	Sengaparile/Lengakapitsi (devil’s claw)	Sun-dried secondary tubers or roots are pulverized into capsules for average preparation	Intestinal infections, kidney infections	Angola	[47]
				Namibia	
				South Africa	
				Mozambique	
				Zambia	
				Zimbabwe	
*Hibiscus calyphyllus*	Motsididi (Wild Hibiscus/Roselle)	Sun-dried flowers are boiled in water and served hot or cold	Stomach infections	West, North, East and some parts of Southern Africa	[42]
*Hirpicium bechuanense*	Kgalemela (S. Moore-Rossler)	Dried tubers or roots are boiled and consumed as tea	Infections and diarrhea	South Africa	[42]
				Mozambique	
				Zimbabwe	
*Mimosopus zeyheri*	Mompudu (red milkwood)	Fruit is consumed fresh or as a dry pulp. Pulverized flowers are used to make snuff. Dried bark, leaves, and roots are boiled and served as tea	Mouth wounds, Dental infections, STIs, soft tissue infections	Eswatini	[74]
				Mozambique	
				South Africa	
*Myrothamnus flabellifolius*	Kgomodimetsing (resurrection plant)	Leaves are pulverized to make a hot beverage	Respiratory infections, soft tissue infections	Angola	[75]
				DRC	
				Kenya	
				Tanzania	
				Lesotho	
				Malawi	
				Mozambique	
				Namibia	
				South Africa	
				Zambia	
				Zimbabwe	
*Ozoroa paniculosa*	Monokane (bushveld ozoroa/common resin tree)	Sun-dried bark, stems, leaves, or roots are pulverized for beverage preparation	Fevers associated with infections	Namibia	[47,76]
				South Africa	
				Zimbabwe	
*Sanseviera scabrifolia*	Mosokelatsebeng (bowstring hemp)	Fleshy leaves are warmed in a fire and the juice is squeezed into the ear	Ear infections	Namibia	[53,67]
				Mozambique	
				South Africa	
				Zambia	
				Zimbabwe	
*Scadoxus puniceus*	Mathubadifhala (blood lily)	Dried tubers or roots are boiled and served as tea or applied on affected areas	Wound infections	Eswatini	[77]
				Ethiopia	
				Lesotho	
				Malawi	
				Mozambique	
				South Africa	
				Tanzania	
				Zambia	
				Zimbabwe	
*Sclerocarya birrea*	Morula/Marula	Fruit is eaten as is Stem or bark is boiled in water and served hot or cold. Seeds are crushed to extract the oil for skin application	Skin infections Mouth wounds	Angola	[47]
				Eswatini	
				Malawi	
				Mozambique	
				Namibia	
				South Africa	
				Zambia	
				Zimbabwe	
*Securidosa longipenduculata*	Mmaba (violet tree)	Dried stem bark and roots are boiled and orally administered. Pulverized bark makes a paste and is applied to affected areas	Skin infections, respiratory infections, urinary tract infections	Tropical and Subtropical Africa	[78]
*Terminalia sericea*	Mogonono (silver cluster-leaf)	Leaves and roots are boiled and consumed as tea. Pulverized leaves are applied as a paste on affected area	Respiratory infections, intestinal infections, soft tissue wounds	Angola	[79]
				DRC	
				Mozambique	
				Namibia	
				South Africa	
				Tanzania	
				Zimbabwe	
*Vanguera infausta*	Mmilo (wild medlar)	Fruit is eaten fresh or dry. Seeds are eaten as they are. Root and leaves are boiled and orally administered	Mouth wounds, respiratory infections, intestinal infections	Kenya	[80]
				Madagascar	
				Malawi	
				Mozambique	
				Namibia	
				South Africa	
				Tanzania	
				Uganda	
				Zimbabwe	
*Ximenia americana/Ximenia caffra*	Moretologa wa pudi/moretologa wa kgomo	Dried and crushed leaves are boiled in water for oral administration. The water extracts are applied on direct sores. The fruit is eaten as is	Body sores, STIs, diarrhea, sore eyes	Angola	[54]
				Lesotho	
				Malawi	
				Mozambique	
				South Africa	
				Swaziland	
				Tanzania	

### 3.3. Plant Secondary Metabolites with Antimicrobial Potential

Plants are known to synthesize and produce diverse groups of organic compounds that are involved in assorted metabolically related functions of the plant, known as secondary metabolites [81]. Their primary function to the plants is in the interaction of the plant with the environment, and they are mostly released in response to abiotic and biotic stresses, thereby supporting plant survival as molecules of defenses [82]. Examples of classes of secondary metabolites include terpenoids, phenols, and derivatives, as well as glucosinolates and alkaloids [83,84].

#### 3.3.1. Alkaloids

Alkaloids are nitrogenous compounds that can be classified as natural, semi-synthetic, and synthetic or based on their chemical structure into typical alkaloids with a heterocyclic ring or atypical alkaloid non-heterocyclic ring [5]. Additionally, they may be split into several classes: tropanes, indole, purines, imidazole, pyrrolidine, pyrrolizidine, isoquinoline, piperidine, and quinolizidine [85]. The antibacterial capacity of alkaloids has been documented and has been linked to efflux pump inhibition, bacterial cell wall synthesis inhibition, changes in cell membrane permeability, inhibition of bacterial metabolism, and nucleic acid and protein synthesis [85,86]. For example, strychnine from *C. tomentosa* has antibacterial activity against *E. coli*, *P. aeruginosa*, and *K. pneumoniae* (Table 3). Other plant-associated alkaloids include nicotine, ephedrine, morphine, and quinine (Figure 2).

#### 3.3.2. Polyphenols

Polyphenols are a large group of secondary metabolites that are classified according to their phenolic groups and structural elements as flavonoids, stilbenes, lignans, tannins, and phenolic acids [87]. The antibacterial capacity levels of polyphenols towards Gram-negative and positive MDR bacteria have been linked to their ability to bind to bacterial enzymes via a hydrogen bond, inducing several modifications in cell membrane permeability and cell wall integrity [86,88,89]. Examples include catechin from *Adansonia digitata* and vitexin from tannic acid from *Dichrostachys cinerea* (see Table 3). Tannin is a descriptive name for a group of polymeric phenolic substances capable of tanning leather or precipitating gelatin from solution [57].

#### 3.3.3. Terpenes

Terpenes are a large group of hydrocarbons synthesized from the 5-carbon precursor units of isopentenyl pyrophosphate and its functional isomer dimethylallyl pyprophosphate [90]. According to the number of isoprenes, they are classified into monoterpenes (e.g., limonene from *A. afra* (Table 3)), diterpenes (e.g., retinol), triterpenes (e.g., oleanolic acid from the *Hibiscus* spp.), and tetraterpenes (e.g., lutein, brassicasterol, campesterol, and β-sitosterol from *Artemisia)* [35,91,92] (Figure 3). The broad antibacterial activity of the terpenes includes efflux pump inhibition and the inhibition of bacterial growth and membrane properties towards MDR bacteria *E. coli*, *S. aureus*, and *Enterobacter* species [91,93,94] (see Table 3).

**Table 3 biomedicines-11-02605-t003:** Studied secondary metabolites from selected African medicinal plants with antimicrobial properties against MDR bacteria.

Plant	Part	Active Compound	MDR Bacteria Active Against	Reference
*Adansonia digitata*	Leaves, stems, roots	Rutin Catechin Gallic acid Caffeic acid	*E. coli* *E. aerogenes* *K. pneumoniae* *P. aeruginosa* *S. aureus*	[95]
*Aloe zebrina Baker/Aloe ferox*	Leaves, roots, and stems	Stearic acid Palmitic acid	*E. coli* *K. pneumoniae* *P. aeruginosa* *S. aureus*	[96]
*Asparagus africanus*	Roots	Gallic acid Hydroxybenzoic acid Hydroxycinnamic acids	*E. coli* *P. aeruginosa* *A. baumannii*	[97]
*Artemisia afra*	Leaves, roots, and stems	Luteolin β-sitosterol Apigenin Myo-inositol	*E. faecalis* *E. coli* *K. pneumoniae* *P. aeruginosa* *S. aureus*	[98]
*Boscia albitrunca*	Roots, leaves	Anthraquinones, martynoside	*E. coli* *K. pneumoniae* *S. aureus*	[69,99]
*Capparis tomentosa*	Roots, fruit	Stachdrine Betane Strychnine 3-hydroxy-4methoxy-3-methyl-oxindole	*E. coli* *P. aeruginosa* *K. pneumoniae*	[100]
*Cassia abbreviata*	Bark	Spectaline Iso-6-cassine	*E. coli* *K. pneumoniae* *P. aeruginosa*	[101]
*Colophospermum mopane*	Seeds, husks, leaves	Labdane, isolabdane, and clerodane diterpenoids	*E. coli* *S. aureus* *Enterococcus*	[102]
*Combretum hereroense*	Bark, leaves, roots	Vitexin, saponaretin, combretacin	*E. coli* *S. aureus*	[103]
*Commiphora glandulosa*	Resin	1β,2β,3β-trihyddoxy-urs-12-ene-23-oic-rhamnoside	*E. coli* *K. aerogenes* *P. aeruginosa* *S. aureus*	[70]
*Dichrostachys cinerea*	Roots	Tanninic acid	*S. aureus* *E. coli* *P. aeruginosa*	[57]
*Dicoma anomala Sond.*	Aerial parts of the plant	Gemacrene, β-farnasene, α-humulene	*E. coli* *S. aureus* *P. aeruginosa*	[104]
*Elephantorhiza goetzei*	Rhizomes, roots	Tanninc acid	*S. aureus* *P. aeruginosa*	[72]
*Euclea undulata*	Bark, leaves, roots	Diopyrin Lupeol 7-methyljuglone	*E. coli* *P. aeruginosa*	[62]
*Hibiscus calyphyllus*	Aerial parts	Oleanolic acid β-amyrin	*S. aureus* *P. aeruginosa*	[40]
*Myrothamnus flabellifolius*	Leaves, twigs	Camphor 1,8-cineole α-pinene	*K. pneumoniae* *S. aureus*	[105,106]
*Ozoroa paniculosa*	Leaves	Anacardic acid Quercetin Proanthocyanidin Gallotannin	*E. coli* *S. aureus* *P. aeruginosa* *E. faecalis*	[61]
*Scadoxus puniceus*	Bulbs, roots, leaves, stems	Haemanthamine Metolachlor	*E. coli* *K. pneumoniae* *S. aureus*	[107]
*Terminalia sericea*	Roots, stems, bark, leaves	Hydroxystilbene glycoside, triterpene sericoside, resveratrol-3-*O-β-D*-rutinoside, Lupeol, Anolignan B	*E. coli* *K. pneumoniae* *P. aeruginosa*	[64,66]
*Ximenia americana/Ximenia caffra*	Bark, fruit, roots, seeds, leaves	Catechin Gallic acid Quercetin Proanthocyanidin	*E. coli* *P. aeruginosa*	[53,55]

### 3.4. Therapeutic Potential of Traditional Medicinal Plants against MDR Bacteria

Antibiotics are definitely the cornerstones of modern medicine; unfortunately, we are still experiencing the rapid spread of foodborne pathogens, emergence of antibiotic-resistant microbial strains, and increasing failure of available chemotherapeutics [108]. Hence, there is a need for efficient antibacterial agents to fill the gap for the discovery and development of anti-MDR agents. Natural products dominate the preferred chemical scaffolds for the discovery of antibacterial agents [13]. In fact, in Africa, traditional healers, herbalists, and individuals have used a variety of wild leaves, roots, barks, fruits, and seeds to combat miscellaneous bacterial illnesses and diseases [109]. Medicinal plants have a wide array of phytochemicals, alkaloids, phenolics, polyphenols, flavonoids, quinones, tannins, coumarins, terpenes, lectins, and saponin, which have been studied due to their mechanisms of action against drug-resistant pathogenic bacteria [2,110,111]. These plants can be proactively bactericidal or bacteriostatic because they have less nitrogen, sulfur, phosphorus, and halogens, and exhibit overall enhanced scaffold variety, molecular complexity, stereochemical abundance, diversity in the ring system, and carbohydrate contents [112,113]. These features allow plant products to modify or inhibit protein interactions, thereby presenting themselves as effective modulators of immune response, mitosis, apoptosis, and signal transduction [112,114]. In this way, the microbial cell can be affected in several ways, including the inhibition and disruption of cell membrane and cell wall functions and structures, interruption of nucleic acid replication and ATP synthesis, generation of reactive oxygen species, inhibition of the formation of biofilms, disruption of quorum sensing at all stages, and via synergy with other antimicrobial agents [94,115,116,117,118,119,120,121].

#### Antibacterial Activity of Plant-Derived Bioactive Compounds

Medicinal plants associated with low molecular weight antibiotics are classified into two types, phytoanticipins and phytoalexins [122]. Phytoanticipins are involved in microbial inhibitory actions while phytoalexins are generally antioxidative and are synthesized de novo by plants in response to microbial infections [122,123]. Many studies have shown that the antimicrobial activity of the plant-derived active compounds, namely alkaloids, phenylpropanoids, and terpenoids, have different capacities to promote cell wall disruption and lysis, induce reactive oxygen species production, inhibit biofilm formation, inhibit cell wall construction, inhibit microbial DNA replication, inhibit energy synthesis, inhibit bacterial toxins from effecting the hosts, and even prevent synergetic resistance to antibiotics [120,124,125,126,127,128].

##### Inhibition and Disruption of Cell Wall Construction

The cell wall is the principal stress-bearing and shape-maintaining element in bacteria. Cell walls can be broadly classified into two polyphyletic groups, Gram-positive and Gram-negative, each arising from structural differences linked to the bacterial interactions with the environment [129,130]. Gram-negative cell walls have an additional membrane covered in lipopolysaccharides, while Gram-positive walls contain only one membrane and have a thicker layer of peptidoglycan containing negatively charged teichoic acids [129]. The biosynthesis of bacterial cell wall peptidoglycan, also called murein, is a complex process that involves enzyme reactions that take place in the cytoplasm on the inner side and outer side of the cytoplasmic membrane [130,131]. Teichoic acid is synthesized as a lipid-linked precursor in the cytoplasm, translocated across the cytoplasmic membrane, and then covalently bonded to the peptidoglycan by the involvement of specific enzymes, namely TagF, TagG, and TagH [130,132]. Since the bacterial cell wall is responsible for many metabolic activities, both directly and indirectly, the integrity of the membrane is of paramount importance, and its disruption can lead to metabolic dysfunctions and ultimately bacterial lysis [129,130,132]

Phenolic compounds and quinones, which are aromatic compounds, are usually targeted to microbial cell surface adhesins and membrane-bound polypeptides, which may disrupt the development of the cell wall and lead to eventual cell lysis [116]. Catechins attract lipid bilayers of the membrane; they form hydrogen bonds, which attract polar head groups of lipids at the membrane edge, which causes structural changes in the cell membrane [133]. Phenolic compounds consist of a hydroxyl functional group, which leads to reactions of hydroxylation, while flavonoids, in particular, are able to complex with bacterial cell walls and disrupt microbial membranes [116,117]. Flavonoids, quercetin, rutin, naringenin, sophoraflavanone, chalcone, and tiliroside are linked to decreased lipid bilayer thickness and fluidity levels and increased membrane permeability and eventual leakage of intracellular proteins and ions in *S*. *aureus* and *S. mutans* [11,134]. Other active flavonoids, including acacetin, apigenin, morin, and rhamnetin, are known to cause the disarrangement and disorientation of the lipid bilayer, thereby weakening the bacterial cell wall and ultimately causing vesicle leakage [125,135]. Flavonoids such as EGCG, penta-hydroxy-flavones, and dimethoxyflavones inhibit the malonyl CoA-acyl carrier protein transacylase, which regulates bacterial fatty acid synthase-II [136,137].

##### Inhibition of Bacterial DNA Replication and ATP Synthesis

Plant-derived bioactive compounds have been proven to halt many enzymes that are linked, directly and indirectly, to DNA replication, transcription, and translation [138].

The replication of DNA is an ATP-dependent process, and the initiation of DNA replication thereof is linked to the growth-dependent accumulation of the ATP-bound form of the highly conserved protein DnaA [139]. As the bacterial cell grows and develops until it reaches a specified mass or size, DnaA-ATP accumulates. Adenosines, ATPase, and DnaA initiate DNA replication by mediating the unwinding of an ATP-rich stretch of DNA within the origin, facilitating the addition of the replication machinery [139]. Flavonoids such as baicalein, morin, quercetin, and silymarin can constrain the function of ATPases, which is the hydrolysis of a phosphate bond in ATP to generate energy, and can obstruct ATP synthesis [119,140]. This obstruction of the synthesis of ATP can disrupt the unwinding of DNA and halt DNA replication. Additionally, EGCG, flavones, and proanthocyanidins have been implicated in the inhibition of the enzymatic activity of ATPase in some bacteria [136,141]. Apigenin, genistein, and myricetin have been recognized as DNA gyrase inhibitors [138,142]. They have been proven to intercalate with the stacking of nucleic acid bases, as well as binding to the β subunit of gyrase and the corresponding blockage of the ATP binding pocket to halt DNA and RNA [122,143]. Helicase is responsible for unzipping the DNA double strand during the early stages of DNA replication [144]. Luteolin, morin, and myricetin have been demonstrated to inhibit the DNA-separating and -rearranging capacity of the helicases of *E. coli* [145,146].

Berberine and harmane, which are alkaloids, have been reported to intercalate with bacterial DNA and lead to the impairment of the cell membrane, consequently leading to cell death [147]. In another study, *E. coli* and *L. monocytogenes* were treated with trans cinnamaldehyde, an essential oil produced by cinnamon, and it was found that it can downregulate F1F0-ATPase, resulting in rapid reductions in ATP, thereby preventing the concentration of cellular ATP and reducing the growth of the bacterial cell [148]. According to Almuhayawi (2020) [149], the ATPase activity in *E. coli* was also inhibited as a result of quercetin binding to the bacterial DNA gyrase B subunit. The disruption of ATP synthesis and DNA replication in bacterial cells can lead to growth disturbances, a compromised cellular structure, and reduced resistance against conditions that may cause cell destruction and ultimate lysis [148,149].

##### Disruption of Pathogenicity Activities

The pathophysiology of a microbial infection in a host is mediated by multiple virulence factors, which are expressed at different stages of infection to cause the disease. These factors include capsule production, toxin production, and hydrolytic protein production; hence, virulence factors are the prime targets for therapeutic interventions and vaccine development [150]. Microbial toxins include exotoxins (secreted by bacteria) and endotoxins (released after bacterial lysis). Many phenolic compounds can affect enterotoxin production through several modes of action, including translation or transcription inhibition, the disruption of secretory mechanisms, the inhibition of quorum sensing regulatory systems, and toxin inactivation or neutralization [150,151,152]. RG tannin and apple phenols were reported to inhibit ADP–ribosyltranferase activity that is critical for cholera toxin action, leading to reductions in toxin-induced accumulation in mouse ileal loops [153,154]. Toxin-mediated pathogenesis is more aggressive because even post cell lysis the toxin remains behind and continues virulence [155]. Therefore, compounds that do not only cause cell death but also inactivate toxins are more effective against this kind of virulence. Phenolic hydroxylic compounds can inactivate or neutralize bacterial exo-proteins by forming hydrogen bonds with active sites of enzymes or toxins to inhibit their activity [151,156]. In some studies, resveratrol disrupted the toxin’s internalization and activity, epigallocatechin gallate and procyanidin blocked the toxin binding and occupied the binding sites, and kaempferol and quercitrin could directly inhibit the activity of the catalytic subunit [156,157]. Gallotannins have inhibitory effects on microorganisms. These effects are attributed to their complexing properties and ability to interact with proteins and inhibit enzyme activity via their chelation of the metal ions of the active sites [158].

##### Generation of Reactive Oxygen Species

Reactive oxygen species (ROS) and reactive nitrogen species (RNS) can be formed from the partial reduction of molecular oxygen and nitrogen through many continuous metabolic pathways, involving both enzyme-catalyzed and non-enzymatic reactions [159,160,161]. ROS can attack diverse targets to exert antimicrobial activity, which helps to account for their versatility in mediating host defense against a vast range of disease-causing pathogenic organisms [160,161]. In fact, it is the host’s NADPH-dependent NOX2 phagocyte oxidase complex in the immune system that is responsible for the generation of ROS [160]. Therefore, many phenols that have the capacity to dabble between pro-oxidant and antioxidant activities because of their hydroxyl groups can be easily coupled with the human’s defense systems against pathogens [162,163]. Examples of these include quercetin and rutin; with their hydroxyl groups they carry out hydroxylation with the lipid bilayer of the bacteria and prevent the generation of ROS [11,116,117]. Upon penetrating the inner cell membrane, phenols can be oxidized by ROS, converting them into pro-oxidants, which are able to oxidize lipids, proteins, and DNA, leading to bacterial cell lysis [163].

ROS readily attack the polyunsaturated fatty acids of the fatty acid membrane to initiate lipid peroxidation, which compromises the integrity of the cell membrane and facilitates eventual cell lysis [164]. Some ROS include superoxide, hydrogen peroxide, hydroxyl radical, alkoxyl radicals, and singlet oxygen. They have different kinetics and levels of activity [162]. Hydrogen peroxide is a covalent and uncharged molecule that readily mixes with water, and is treated as such by cells, meaning it diffuses across the cell membrane with ease [161]. It can also form redox reactions and highly reactive free radicals, which may be aggressively oxidative and can impair the functions of many metabolic processes [161].

In this manner, plant-derived compounds with hydroxyl tails or double bonds can be oxidized to become phenoxyl radicals or quinone intermediates, meaning they behave as pro-oxidant systems [120,164]. In the presence of transition metals, they can fragment DNA and lead to mutation and cell death [165]. Catechins, for instance, enhance the production of oxidative stress as ROS and RNS [166]. In this way, they can cause altered membrane permeability, bond to the highly negatively charged lipids and liposomes, and facilitate cell damage and eventually death [167]. They are also linked to the leakage of intracellular proteins and ions, as they also disturb the membrane transport system [164,166,167]. Epigallocatechin gallate, in high concentrations, exerts its mode of action of killing bacteria via the generation of ROS, causing alterations of the membrane permeability and membrane damage [120].

##### Formation of Biofilms Inhibition

Biofilm formation is a process whereby microorganisms irreversibly attach to and grow on a surface and produce extracellular polymers that facilitate attachment and matrix formation, resulting in alterations of the phenotype of the organisms with respect to the growth rate and gene transcription [168]. These surfaces of attachment may take many forms, including the surface of the host that is colonized and infected [168,169]. The presence of mobility structures such as flagella, fimbriae, and pili is most important for microbial attachment [168,169]. Microbial cells, via these appendages, temporarily attach to the surface, as they prepare to form colonies and exude extracellular polymers that will then attach permanently to the surface [170,171].

Flavonoids have been described to cause the aggregation of multicellular composites of bacteria and cause psuedobacterial growth after aggregation, which indicates that flavonoids are potent antibiofilm compounds [122]. Flavonoids such isovitexin, galangin, EGCG, trihydroxyflavanols, and 3-O-octanoyl-epicatechin have been shown to inhibit the multicellular aggregation of pathogenic bacteria [122,124,127,172]. Moreover, chrysin, phloretin, naringenin, epicatechin gallate, and proanthocyanidins have been proven to inhibit N-acyl homoserine lactone-mediated quorum sensing [173,174,175]. The synthetic flavonoid lipophilic-3-arylidene was found to be very active against *S. aureus*, *S. epidermidis*, and *E. faecalis* due to a bacterial cell clump that influences the integrity of the cell wall as a result of biofilm disruption [155]. Phloretin was also proven to prevent the formation of fimbriae in *E. coli* by reducing the expression of the curli genes (*csg*A, *csg*B) and toxin genes (hemolysin E, Shiga toxin), eventually halting the formation of the biofilm [176,177].

In one study, the components of cranberry juice were hypothesized to interact with hydrophobic proteins on the surface of the bacterial cell [178]. *Streptococcus mutans* and *Streptococcus sobrinus* were treated with cranberry juice, and it was found that the cranberry juice reduced the surface hydrophobicity of the cells, interfering with adhesion and the initial stages of biofilm formation. Since hydrophobicity is an important factor in the initial attachment of the bacteria to a surface, reducing the hydrophobicity decreases the likelihood of adhesion [179]. Therefore, preventing biofilm formation is a potential method of curbing bacterial colonization and eventual pathogenicity, possibly through the engagement of plant-derived bioactive compounds.

##### Quorum Sensing (QS) Inhibition

The discovery of bacterial communication systems (QSsystems) that orchestrate important temporal events during the infection process has afforded a novel opportunity to ameliorate bacterial infections using compounds and possibly overriding bacterial signaling mechanisms [180]. Many plant species produce secondary metabolites and compounds that can control the growth of microbes and have traditionally been used to treat human microbial infections via anti-quorum sensing. The therapeutic substances in *Centella asiatica* L. Urban are saponin-containing triterpene acids and their sugar esters, e.g., Asiatic acid, madecassic acid, and asiaticosides, which have been shown to treat skin problems and heal wounds [181,182]. In fact, in a study by [183], the anti-quorum sensing potential of *C. asiatica* was investigated using *Chromobacterium violaceum* and *P. aeruginosa* PA01. The anti-quorum sensing activity of an ethanolic extract and ethyl acetate fraction of the herb were investigated against *C. violaceum*. The ethanol extract showed quorum sensing inhibition against violacein production, while the ethyl acetate fraction was four times more active against violacein production, without significantly affecting the growth [183]. The ethyl acetate fraction also inhibited quorum-sensing-regulated phenotypes, pyocyanin production, elastolytic and proteolytic activities, swarming motility, and biofilm formation in *P. aeruginosa* PA01 in a concentration-dependent manner [183].

Bacteria are consistently subjected to environmental stimuli such as nutrient availability, temperature, and pH changes. They in turn have developed multiple systems that form cell-to-cell communication via the use of autoinducers of N-acyl-homoserine lactone (AHLs) for Gram-negative bacteria and oligopeptides for Gram-positive bacteria in order to activate specific gene expression and QS systems [184,185,186]. The AHL-mediated QS function requires three major components: an AHL signal molecule, an AHL synthase protein to make the AHL signal (LuxI), and a regulatory protein that responds to the surrounding concentration of AHL [187,188]. At high AHL concentrations, the sigma receptor protein (LuxR) forms a complex with AHL and becomes activated; this activation triggers the expression of genes that are linked to biofilm formation and virulence during pathogenesis [185,187].

The traditional treatment of bacterial infections is anchored on compounds that either kill or inhibit bacterial growth [189]. With the development of resistance against and tolerance for these antibacterial agents, hacking into the quorum sensing systems using plant-derived substances may pose a potential novel opportunity to amend bacterial virulence [190,191]. This is because many compounds that have been proven to override bacterial signaling are present in medicinal plants, and bacterial communication systems are responsible for the orchestration of the important progressive events during the infection process [186,188,190,191]. Therefore, the strategies aiming to interrupt bacterial quorum sensing circuits possibly include the inhibition of AHL or oligopeptide signal generation, inhibition of AHL or oligopeptide signal distribution by antagonization or degradation, and prevention of AHL signal reception via competitive and non-competitive inhibition [184,190]. The plant-derived molecules that affect these AHL-mediated quorum sensing stages include sulfur-containing compounds, monoterpenes, monoterpenoids, phenylpropanoids, benzoic acid derivatives, diarylheptanoids, coumarins, flavonoids, and tannins [189,192].

##### Inhibition of AHL or Oligopeptide Signal Generation

The AHL signal is synthesized at a low basal level by the AHL synthase. Often the quorum sensor is subject to autoinduction because the gene encoding the signal synthase is among the target genes; hence, a positive feedback regulatory loop is created [189,192]. The autoinduction allows a rapid increase in signal production and dissemination, which in turn induces a quorum-sensing-controlled phenotype throughout the bacterial population [189]. AI-1 is an AHL that is involved in intraspecies communication, whereas AI-2, a furanosyl borate diester, has been suggested to play an important role in interspecies communication [193]. Genome sequencing revealed the presence of *luxS* homologs (encoding the AI-2 signal synthase) in many pathogens, including *E. coli*, *Helibacter*, *Neisseria*, *Porphyromonas*, *Proteus*, *Salmonella*, *E. faecalis*, *S. pyogenes*, and *S. aureus*, and in some of the pathogens *luxS* is required for virulence [194,195]. The luxS gene responsible for the production of AI-2 is produced from S-ribosylhomocysteine, a product in S-adenosylmethione utilization [194,196]. LuxS cleaves S-ribosylhomocysteine to form homocysteine and 4,5-dihydroxy-2,3-pentanedione, which is subsequently converted to AI-2 through an unknown biochemical pathway [193,197]. LasI, which is part of one of the quorum sensing systems in *P. aeruginosa (*LasR/I and RhII/R), is essential for the production of the AHL molecule, N-(3oxododecanoyl)-L-homoserine lactone (3O-C12-HSL), and lasR is the transcriptional regulator [118,198]. The second QS system comprises the RhII and RhIR proteins [118,198]. The RhII synthase mediates the synthesis of the signal molecule, N-butyryl-homoserine lactone C4-HSL, while RhIR is the transcriptional regulator [198,199].

The majority of bacteria that produce AHL signals encode one or more genes homologous to luxI of *Vibrio fischeri* [189]. The expression of these genes in a heterologous host background has demonstrated that the LuxI-type protein is required and sufficient for the production of AHL signals. Additionally, the catalysis of AHL synthesis involves a sequentially ordered reaction mechanism that uses S-adenosyl methionine (SAM) as the amino donor for the generation of the homoserine lactone ring moiety and an appropriately charged acyl carrier protein (ACP) as the precursor for the acyl side chain of the AHL signal [200]. Various analogs of SAM, such as S-adenosylcysteine and sinefungin, have been demonstrated to be potent inhibitors of AHL synthesis catalyzed by the *P. aeruginosa* RhII protein [189]. Moreover, such inhibition can be due to interference with acyl homesrine lactone production (*luxI* effect) or a transcription response (*luxR* effect) [191]. Protein inhibition at the ribosomal level can be a means of hacking into the quorum sensing system of the pathogen.

*Trans*-cinnamaldehyde was reported to reduce the expression of *luxR*, which codes for the transcriptional regulator of quorum sensing in *C. sakazakii* [201]. Similarly, garlic extract and p-coumaric acid inhibited quorum sensing in quorum sensing strains, indicating that plant compounds potentially modulate virulence by affecting quorum sensing in microbes [150]. In one study, *S. aromaticum* exhibited high quorum sensing inhibition ability and concentration-dependent violacein pigment inhibition of *C. violaceum*, which is also correlated with biofilm formation inhibition [202,203]. The inhibition of biofilm formation indicated that EPS was not formed and that the bacteria were unable to attach to the host for the formation of biofilms [204].

Decreasing the active signal molecule concentration in the environment can inhibit bacterial cell-to-cell communication. The reversible hydrolysis of AHLs at high pH levels has been recorded in some instances and was linked purely to alkaline conditions [205]. In other cases, the degradation of the AHLs was linked to enzymatic activity. *Bacillus* species produce an enzyme, termed AiiA, which catalyzes the hydrolysis of AHL molecules. The expression of this gene in the plant pathogen *Erwinia carotovora* resulted in the reduced release of AHL signals, decreased extracellular pectolytic enzyme activity, and attenuated soft rot disease symptoms in all plants tested [206]. Moreover, transgenic plants expressing AiiA have been shown to be significantly less susceptible to infection by *E. carotovora* [168]. Therefore, AHL-degrading or antagonizing proteins are of great clinical potential for use in the prevention of diseases caused by quorum-sensing-proficient bacterial populations.

Blocking of the quorum sensing signal transduction can be achieved by an antagonist molecule capable of competing or interfering with the native AHL signal for binding to the LuxR-type receptor [180,192]. Competitive inhibitors share structural similarities with the native AHL signal; in that way they bind and occupy the AHL binding site and block the LuxR-type receptor [180,189,207]. The non-competitive inhibitors, on the other hand, show no or little structural similarities to AHL signals, as these molecules bind to different sites on the receptor protein. In this manner, it would be more effective to locate competitive inhibitors. AHLs have acyl side chains, whose length and flexibility are also crucial in the effective binding of the AHL to the LuxR-type receptor [188,208]. Receptors also have ion ends that are respective to their partners when coupling.

Several plants secrete bioactive compounds that mimic bacterial AHL signal activities and affect quorum-sensing-regulated behaviors in associated pathogens [209,210]. Quercetin, among other phenols, was implicated to bind the LuxR-type receptor proteins (LasR, RhIR, and CviR) and inhibit quorum-sensing-regulated bioluminescence in *V. harveyi* [211]. The Australian red macroalga *D. pulchra*, *Hoslundia opposita*, and *Lippia javanica* produce ranges of furanone compounds that display antifouling and antimicrobial properties [212,213]. Furanones were implicated in the displacement of radiolabeled AHL molecules from LuxR, suggesting competitive inhibition by binding on the LuxR receptor site [180,207]. Moreover, these furanones have been linked to the inhibition of AHL-controlled virulence factor production and pathogenesis in *P aeruginosa*. The furanones are also linked to the inhibition of the quorum-sensing-controlled luminescence and virulence of the black tiger prawn pathogen *Vibrio harveyi*, as well as the inhibition of the quorum-sensing-controlled virulence of *E. carotovora* [189,192,207,214]. Anti-quorum-sensing activity against *S. aureus* was also linked to *L. javanica*-derived limonene, carvone, geranial, and neral essential oils [189,192,207,209,214]. Other flavonoids such as apigenin, kaempferol, naringenin, and quercetin were also proven to possess anti-quorum-sensing capacities by inhibiting enteroaggregative biofilm formation in a concentration-dependent manner [215,216].

##### Synergy with Other Antimicrobial Agents

The synergistic behavior of traditional antibiotics with compounds obtained from plants offers many advantages, such as increased efficiency, decreased undesirable effects, increased stability or bioavailability of free agents, and the ability to obtain suitable therapeutic effects at lower doses [217]. Many studies have demonstrated that the synergism between different antibacterial agents can yield better effects than merely one. This can include the combined effects of plant-derived compounds alone or plant-derived compounds with antibiotics. Mixing plant extracts with different commercial antibiotics enhanced and synergized their antibacterial effects [218]. Additionally, the synergic effects between different single compounds could trigger the antimicrobial effectiveness of the plant-derived compounds and may reduce the resistance of many pathogenic microorganisms.

In one study, anti-bacterial activity against *Listeria monocytogenes* biofilm was more effective where the synergism between a-pinene, limonene, and linalool was at play than when each single component was used [219]. Carvacrol (Table 1), γ-terpinene, and ρ-cymene can be effective and have synergistic effects when combined; this is due to the action of ρ-cymene, which works as a mediator for the transportation of carvacrol and γ-terpinene across the cell walls and cytoplasmic membranes of pathogenic microorganisms [220]. In another study, plant extracts from *Mezoneuron benthamianum*, *Securinega virosa*, and *Microglossa pyrifolia* increased the susceptibility of major drug-resistant bacteria such as *S. aureus* and *P. aeruginosa* to antibiotic norfloxacin [221]. Similarly, geraniol, extracted from *Helichrysum italicum*, was found to restore the efficacy of quinolones, chloramphenicol, and β-lactams against multi-drug-resistant pathogens, including *Acinetobacter baumannii* [222]. The flavonoid chalcone can also be used as a therapeutic agent against infections of methicillin-resistant *S. aureus* strains [223]. A bioactive macrocyclic spermine alkaloid from *Albizia schimperiana* was linked to the inhibitory activity against methicillin-resistant *S. aureus* and *E. coli* [224].

Essential oils from different plants demonstrated activities against the emerging multi-drug-resistant bacteria in human skin; this could be used to combat the problem of untreatable skin diseases. In a study by Haroun and Al-Kayali [225], *T. spicata* extracts showed the ability to increase the susceptibility of multi-drug-resistant strains of *S. aureus* and *K. pneumoniae* to various antibiotics. Essential oils and organic extracts of *A. afra*, *A. betulina*, and *S. frutescens*, in combination with ciprofloxacin, also displayed synergistic interactions against *E. coli* [225]. Arasu et al. evaluated the antibacterial activities of essential oils of *Sesamum indicum*, *Allium sativum*, and *Acorus calamus*; they found out that essential oils alone and in combination with antibiotics had high activity levels against some food spoilage bacteria [226]. Indeed, medicinal plants, unlike pharmacological drugs, commonly have several chemicals working together catalytically and synergically to produce a combined effect that surpasses the total activity of the individual constituents [227]. The combined effect of these substances tends to increase the activity of the main medicinal constituents by speeding up or slowing down their assimilation in the body. In one study, plant extracts from *Mezoneuron benthamianum*, *Securinega virosa*, and *Microglossa pyrifolia* increased the susceptibility of major drug-resistant bacteria such as *S. aureus* and *P. aeruginosa* to antibiotic norfloxacin [221]. Similarly, geraniol, extracted from *Helichrysum italicum*, was found to restore the efficacy of quinolones, chloramphenicol, and β-lactams against multi-drug-resistant pathogens, including *Acinetobacter baumannii* [222].

## 4. Conclusions and Future Research

Novel therapeutic options are crucial in combatting AMR and reducing the burden of infectious diseases. Bioactive compounds from indigenous medicinal plants alone or in combination with existing antimicrobials offer promising solutions towards overcoming global AMR threats. As already shown in this review, the African continent is rich in diverse medicinal plant species, and the current research should focus on new discoveries of plant secondary metabolites with antimicrobial properties. There have already been advances in the development of new antimicrobial therapies, which are currently supported by technological advancements in “omics” techniques such as proteomics and metabolomics. Microbial genomics has contributed globally to the understanding of bacterial resistance mechanisms, evolution, and dissemination, allowing better treatment response, prevention, and control strategies. However, Africa is still lagging behind in “omics” research due to socioeconomic factors a lack of access to and sustainable use of genomic sequencing technologies. The economic challenges, limited infrastructure for “omics“ research, and shortage of well-trained staff in the African continent have evidently been the downfall of efforts towards combating AMR and the development of new antimicrobial drugs. Despite these challenges, there is promising research in some parts of Africa, such as Botswana, aimed at addressing AMR problems using secondary metabolites derived from medicinal plants. The preliminary results from some of the indigenous plant extracts studied in Botswana revealed great potential when tested against MDR bacteria. Therefore, extensive research in this area of antimicrobial discovery remains critical towards combating AMR in Africa and globally.

## Figures and Tables

**Figure 1 biomedicines-11-02605-f001:**
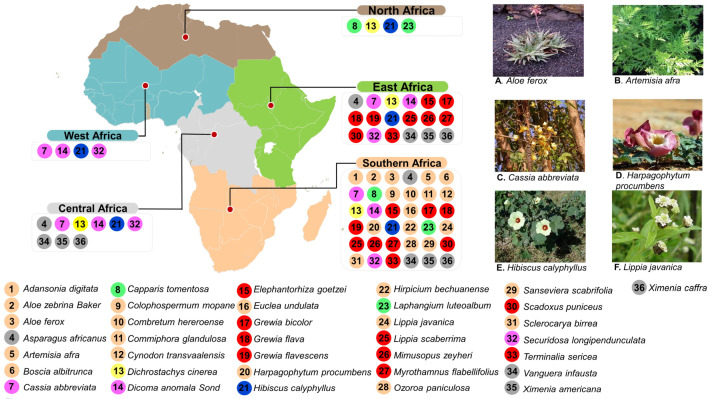
Diverse plant species with potential antimicrobial properties widely distributed across different regions in Africa: (**A**–**F**) examples of plant species that have been commercialized as treatment for bacterial infections. See the Appendix A (Appendix A) for more details about species origins, studies, and sources.

**Figure 2 biomedicines-11-02605-f002:**
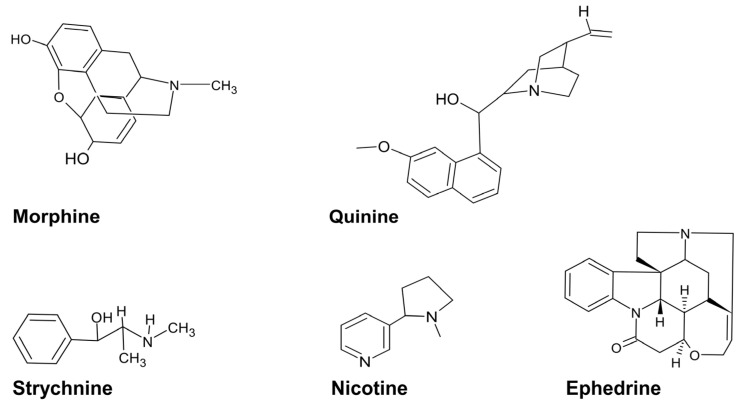
Examples of plant-associated alkaloids.

**Figure 3 biomedicines-11-02605-f003:**
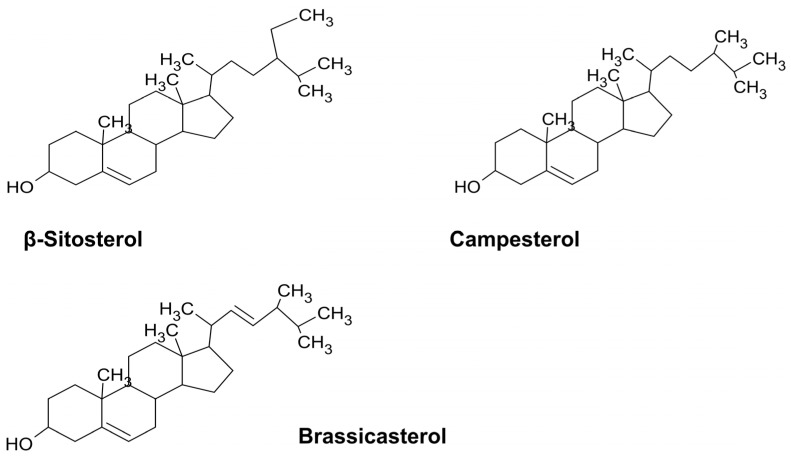
Chemical structures of plant-associated terpenes.

**Table 1 biomedicines-11-02605-t001:** Mechanisms of antibiotic resistance to common classes of antibiotics using healthcare.

Bacterial Strain	Antibiotics	Mechanism of Resistance	Reference
*A. baumanni*	β-Lactams, aminoglycosides, carbapenems	Altered membrane permeability. Enzyme inactivation by aminoglycoside-modifying enzymes	[4,6,8]
*E. coli*	Tetracycline, gentamycin, amoxicillin, trimethoprim	AcrAB-TolC efflux pump	[5,8]
*Enterobacter* species	β-lactams, most antibiotics, carbapenem	Production of inactivation enzymes using extended-spectrum β-lactamases, carbapenemase acetyltransferases, phophotransferases, and adenyltransferase	[13,16]
*K. pneumoniae*	Third-generation cephalosporins, carbapenem, β-lactams	Changes in membrane permeability Alterations of target site Efflux pump Enzyme inactivation by β-lactamases	[9,34]
*S. aureus*	Cephalosporins, methicillin, oxacillin, peniccilin, vancomycin, erythromycin	NorA efflux pumps Mutations in genes involved in cell wall synthesis Alterations of antibiotic binding site	[13,16]
*P. aeruginosa*	Penem groups of antibiotics, β-lactams, fluoroquinolones	Structural changes in the cell’s surface casings Decreases in antibiotic porins MexAB-OprMM efflux pump Enzyme inactivation by β-lactamases Delay of the incoming flow by modifying the structure of the antibiotic	[5,16]

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
