# Peer review of "Potential of Selected African Medicinal Plants as Alternative Therapeutics against Multi-Drug-Resistant Bacteria"

_biomedicines, 2023, doi:10.3390/biomedicines11102605_

Round 1

Reviewer 1 Report

The authors' objective was to showcase promising traditional medicinal plants from Africa and explore their secondary metabolites as viable alternatives for antibiotic therapy against multi-drug resistant bacteria.

The manuscript is highly informative and comprehensive, providing valuable insights. I strongly recommend its acceptance in its current form.

Author Response

Replay to Reviewer 1 is added

Reviewer 2 Report

Dear authors,

 I congratulate you by the good review you have written regarding the interesting topic: African Medicinal Plants.

Some considerations to improve the quality of the manuscript:

The first literature you reference should be papers with the highest quality possible. If you talk about  antibacterial resistance activity in the first paragraph, you should refer to primary scientific literature. Your reference number 2 should be modify.

-Line 35: When you state `potencial aromatic compounds´ you should delete the word `potential´. 

- Is it reference 7 a book? Correct the way you mention this reference, it is not clear.

- Same question for reference 19. Is it a book? Which editorial? Missing information.

- Same question for reference 51.

- Reference 140, some capital letters should be non capital.

- Is the first name in reference 165 correct?

On the other hand, you should up-date the scientific literature introducing some important and recent work. 

Finally, I did not find any figure showing main chemical structures you talk about in the paper. It is better to introduce the main scaffolds in figures in order to get an easier reading.

Current literature regarding antibacterial activity  and other activities from African Plants.

BMC Complementary Medicine and Therapies (2023), 23(1), 100

Virology Journal (2023), 20(1), 50 

Other literature:

Plant Methods (2022), 18(1), 87

Evidence-based complementary and alternative medicine : eCAM (2022), 2022, 8023866 

Scientific Reports (2022), 12(1), 10305

Best regards

It is fine.

Author Response

reply to reviewer 2 is added

Reviewer 3 Report

First of all the title of the manuscript need to be rectified, the author has not studied all the medicinal plants of Africa they have studied only a few, so should write selected medicinal plants in the title. Besides, the data provided is preliminary data and there is only one table. Not suitable for this journal.

Next, where is information on the secondary metabolites and their mode of action in the manuscript? These should be discussed in details.

The author should specify which plant species they have studied in the abstract part.

The author should change the organization of the manuscript which is confusing.

Moderate editing of English language required

Author Response

Reply to reviewer 3

Reviewer 4 Report

The title of this article indicates that multidrug-resistant bacterial strains will be considered. Nowhere in the text is there any mention of any resistant bacterial strain.

The main literature reference reports the biological activities of extracts of various plants growing throughout the African continent. Nowhere in the text is resistance found, much less multi-resistance. These facts give the impression of a mismatch between subject and content.

The same inconsistency is found in the thus formulated topic and purpose of the research.

"The aim of this review is to highlight some promising traditional medicinal plants found in the African continent and provide insights into their secondary metabolites as alternative options in antibiotic therapy against multi-drug resistant bacteria."

Nowhere in the text is a systematized metabolic composition of the mentioned plant found. Moreover, to provide insight into their secondary metabolites as alternative options for antibiotic therapy against multidrug-resistant bacteria.

The chapter "Introduction" contains facts that, apart from being repeated, are far from the topic - the article is not devoted to the analysis of economic data, nor to forecasts of the impoverishment of the world's population.

This review lacks an entire "Materials and Methods" chapter to make it clear which databases were used to collect data for the review article.

My recommendation is that the authors make a plan for the article, in which they foresee such points that correspond to the topic - an introduction from which it is clearly formulated why they have chosen the goal, the goal is clearly and precisely formulated, and they must include a chapter "Materials and methods".

Moderate editing of the English language is required.

Author Response

Reply to reviewer 4

Round 2

Reviewer 3 Report

Still, I feel the title should be revised and the manuscript overall is not suitable for this journal. It is more specific for a plant science-related journal rather than a biomedicine journal.

Minor editing of English language required

Reviewer 4 Report

Please, see the file attached.

The English language must be redacted. 

Author Response

Please see the attached file for ansers

Round 3

Reviewer 3 Report

Still, I feel it is not suitable for this journal. It is more suitable for a plant science-related journal.

Minor editing of English language required

Reviewer 4 Report

Place the table of abbreviations at the end of the article.

Abbreviation terms should be spelled out the first time they are used in the text, then the abbreviation should be placed in parentheses.

Names of bacterial strains should be in italics.

Subpoint names should also be italicized.

3.4.1.4. Generation of Reactive Oxygen Species - it is not clear whether these plants support the generation of ROS or inhibit it.

Is there evidence for extracts from the plants studied to inhibit the generation of ROS?

The same question can be asked about inhibiting the formation of biofilms.

Synergy with other antimicrobial agents - between which substances and in relation to what is synergistic action observed?

In conclusion - current or future research needs to be conducted?

The conclusion contains data that is more relevant to the introduction.

Moderate editing of the English language is required.
